# Bile Duct Ligation Impairs Function and Expression of Mrp1 at Rat Blood–Retinal Barrier via Bilirubin-Induced P38 MAPK Pathway Activations

**DOI:** 10.3390/ijms23147666

**Published:** 2022-07-11

**Authors:** Ping Li, Yiting Yang, Zijin Lin, Shijin Hong, Ling Jiang, Han Zhou, Lu Yang, Liang Zhu, Xiaodong Liu, Li Liu

**Affiliations:** Center of Drug Metabolism and Pharmacokinetics, School of Pharmacy, China Pharmaceutical University, Nanjing 210009, China; 3119010071@stu.cpu.edu.cn (P.L.); 1821010209@stu.cpu.edu.cn (Y.Y.); linzj99@163.com (Z.L.); hsj1826@163.com (S.H.); 3120010078@stu.cpu.edu.cn (L.J.); 15305195230@163.com (H.Z.); yanglu2214@163.com (L.Y.); a8459022@126.com (L.Z.)

**Keywords:** bile duct ligation, blood–retinal barrier, multidrug resistance-associated protein 1, liver injury, p38 MAPK, bilirubin

## Abstract

Liver injury is often associated with hepatic retinopathy, resulting from accumulation of retinal toxins due to blood–retinal barrier (BRB) dysfunction. Retinal pigment epithelium highly expresses MRP1/Mrp1. We aimed to investigate whether liver injury affects the function and expression of retinal Mrp1 using bile duct ligation (BDL) rats. Retinal distributions of fluorescein and 2,4-dinitrophenyl-S-glutathione were used for assessing Mrp1 function. BDL significantly increased distributions of the two substrates and bilirubin, downregulated Mrp1 protein, and upregulated phosphorylation of p38 and MK2 in the retina. BDL neither affected the retinal distribution of FITC-dextran nor expressions of ZO-1 and claudin-5, demonstrating intact BRB integrity. In ARPE-19 cells, BDL rat serum or bilirubin decreased MRP1 expression and enhanced p38 and MK2 phosphorylation. Both inhibiting and silencing p38 significantly reversed the bilirubin- and anisomycin-induced decreases in MRP1 protein. Apparent permeability coefficients of fluorescein in the A-to-B direction (P_app, A-to-B_) across the ARPE-19 monolayer were greater than P_app, B-to-A_. MK571 or bilirubin significantly decreased P_app, A-to-B_ of fluorescein. Bilirubin treatment significantly downregulated Mrp1 function and expression without affecting integrity of BRB and increased bilirubin levels and phosphorylation of p38 and MK2 in rat retina. In conclusion, BDL downregulates the expression and function of retina Mrp1 by activating the p38 MAPK pathway due to increased bilirubin levels.

## 1. Introduction

Retina, being a part of the central nervous system, shares similar anatomical and physiological features with the brain [1]. The links between eye pathology and brain disorders such as Alzheimer’s disease, Parkinson’s disease, and hepatic encephalopathy have been demonstrated [2,3,4]. For example, patients suffering from hepatic failure and accompanying hepatic encephalopathy often display functional abnormalities of the retina, including blue-yellow defects and decreases in visual acuity, defined as hepatic retinopathy [3,5,6]. Functional studies using electroretinogram have revealed that oscillatory potentials and a- and b-wave amplitudes were decreased and delayed in patients with liver failure, whose extents of alterations were associated with the degree of hepatic encephalopathy [5] and liver cirrhosis [7]. Consistently, patients with restored liver function after liver transplantation displayed significantly improved electroretinogram parameters [6]. Animal experiments have exhibited that liver injury induced by both thioacetamide and bile duct ligation (BDL) can lead to hepatic retinopathy [8,9,10].

The blood–retinal barrier (BRB) is composed of the inner BRB and the outer BRB, which are mainly formed by retinal capillary endothelial cells and retinal pigment epithelial (RPE) cells, respectively. BRB functions are to keep retina homeostasis by controlling molecular exchanges between retina and blood and preventing entry of toxic compounds or biological macromolecules to the neural retina. Under normal physiological conditions, maintenance of BRB function is highly dependent on tight junctions and influx/efflux transporters. The identified efflux transporters at the BRB are the main ATP-binding cassette (ABC) transporters, including P-glycoprotein (P-GP/ABCB1), breast cancer resistance protein (BCRP/ABCG2), and multidrug resistance-associated proteins (MRP1/ABCC1, MRP4/ABCC4, and MRP5/ABCC5) [11,12,13,14,15,16,17].

Liver injury has been demonstrated to alter the expressions and functions of ABC transporters at the blood–brain barrier (BBB). For example, liver injury induced by BDL has been reported to suppress function and expression of Bcrp but enhance the function and expression of P-gp at the BBB of rats [18,19,20]. Acute liver injury induced by thioacetamide significantly decreased the expressions and functions of P-gp and Bcrp but increased the expression and function of Mrp2 at the BBB of rats [21,22]. In general, the characteristic of BRB is similar to that of BBB, indicating that liver injury may alter the functions and expressions of these ABC transporters at the BRB. Although P-gp and Bcrp are expressed at the BRB, a previous report showed that increases in retinal uptake indexes of verapamil (1.6-fold), quinidine (1.1-fold), and digoxin (3.7-fold) in the retina of *Abcb1a*^−/−^ rats were remarkably lower than brain uptake parameters of verapamil (8.3-fold), quinidine (12.3-fold), and digoxin (14.2-fold) compared to wild-type rats [23]. Similarly, triple knockout (*Abcb1a/1b*^−/−^, *Abcg2*^−/−^) or coadministration of Bcrp inhibitor elacridar significantly increased the brain distribution of mitoxantrone by about 3-fold in control mice, but the increases did not occur in the retina [24]. These results indicate that the importance of P-gp and Bcrp efflux at the BRB is less than that at the BBB [14]. MRPs are also expressed at the BRB, especially at the outer BRB [14]. MRP1/Mrp1 has been demonstrated to be expressed in mouse RPE [24], human retina/choroid [25], primary human RPE cells, and human RPE cells (ARPE-19 cell line) [13,16,17,26] at the mRNA or protein level. Our pre-experiment also showed the highest expression of *Abcc1* mRNA among the tested ABC transporters (*Abcb1a/1b*, *Abcg2*, and *Abcc1~6*) in the eyecup of rats. High expression of *ABCC1* mRNA was also detected in ARPE-19 cells.

MRP1, a primary active transporter of xenobiotics and their phase II conjugates, also transports some endogenous substrates, including leukotriene C4, estrone sulfate, dehydroepiandrostenedione sulfate, estradiol-17β-glucuronide, bilirubin, and its glucuronides. Some signaling molecules such as prostaglandin A, 15-deoxy-ΔPGJ, sphingosine-1-phosphate, lysophosphatidylinositol, and 4-hydroxy-nonenal-glutathione conjugate are also endogenous substrates of MRP1 [14]. These results indicate that MRP1 plays an indispensable role in physiology and disease processes.

The aims of the study were: (1) to investigate whether liver injury affects the function and expression of Mrp1 at the BRB using BDL rats as an animal model; (2) to screen factors affecting the expression and function of MRP1 using APRE-19 cells as an in vitro outer BRB model. Hyperbilirubinemia is a typical characteristic of BDL-induced liver injury. The final aim was to investigate whether alterations in expression and function of retina MRP1/Mrp1 are attributed to increased levels of unconjugated bilirubin (UCB) and possible mechanism, which was further confirmed using rats treated with bilirubin.

## 2. Results

### 2.1. Physiological and Biochemical Parameters in Sham and BDL Rats

Physiological and serum biochemical parameters were measured on Day 14 following BDL surgery (Table 1). The results showed that compared with Sham rats, BDL rats showed significantly increased liver/body weight ratio, spleen/body weight ratio, as well as increased serum ALT, AST, ALP, total bile acids, total bilirubin, ammonia, and asymmetric dimethylarginine (ADMA), indicating that liver injury was successfully developed by BDL, and the BDL rats characterized hyperbilirubinemia (HB).

### 2.2. Effect of BDL on Function and Expression of Mrp1 at Rat BRB

Retina distributions of fluorescein and 2,4-dinitrophenyl-S-glutathione (DNP-SG), substrates of Mrp1, were calculated to assess the function of retina Mrp1. It is generally accepted that the retinal concentrations of substrates are affected by plasma concentrations [21], and the retina-to-plasma concentration ratios (K_r/p_) of substrates serve as the index of retinal penetration to eliminate the effects of plasma substrate levels on retinal substrate distribution. The results showed that BDL significantly increased fluorescein concentration in the retina (Figure 1a) without affecting plasma fluorescein levels (Figure 1b). Remarkably increased K_r/p_ of fluorescein was found in BDL rats (Figure 1c). Similarly, BDL did not affect DNP-SG concentration in the retina (Figure 1d) but strikingly decreased DNP-SG concentration in plasma (Figure 1e), leading to remarkably higher K_r/p_ of DNP-SG (Figure 1f) compared with Sham rats.

UCB is an endogenous substrate of MRP1/Mrp1. Thus, levels of UCB in plasma and retina were also measured. As expected, BDL significantly increased the concentration of UCB in both plasma (Figure 1g) and retina (Figure 1h). The retina UCB levels were significantly increased from less than the lower limit of quantitation (LLOQ = 0.67 pmol/mg protein) in Sham rats to 2.72 ± 1.49 pmol/mg protein in BDL rats.

The mRNA levels of *Abcc1~6*, *Abcb1a/1b*, and *Abcg2* were detected in eyecups of normal rats. The results showed that *Abcc1* had the highest expression, followed by *Abcc5* (Figure 2a). To investigate whether the increased retina distributions of fluorescein and DNP-SG in BDL rats come from impairment of Mrp1 expression, Mrp1 protein expression in eyecups of BDL and Sham rats was further measured. Western blot showed BDL significantly decreased Mrp1 protein expression in eyecups by approximately 60% (Figure 2b).

The location of Mrp1 in rat retina was detected using double immunofluorescence staining (Figure 2c). Glut1 was used to label retinal blood vessels and RPE [15,27]. As we can see, Glut1 (red) was mainly localized at retinal microvessels and RPE. In RPE, Glut1 protein was evidently detected on both apical and basolateral membranes. Mrp1 protein (green) was mainly colocated with Glut1 at the basolateral side of rat RPE. The apical side of rat RPE and retinal microvessels only showed very weak staining or no staining of Mrp1 protein. Very weak expression of Mrp1 also existed in retinal ganglion cells. This was in line with alterations in total protein levels of retina Mrp1 that BDL also remarkably decreased expression of Mrp1 at the basolateral membrane of rat RPE (Figure 2d,e).

To exclude the possibility that the increased retina distributions of the substrates result from the BRB breakdown, the integrity of BRB was measured using leakage of fluorescein isothiocyanate-conjugated dextran (FITC-dextran). It was found that although BDL rats showed a slight decrease in plasma FITC-dextran concentration (Figure 2f), retina concentrations of FITC-dextran and K_r/p_ of FITC-dextran in the retina of BDL were comparable to those of Sham rats (Figure 2g,h). Consistently, BDL barely affected expressions of tight junction proteins ZO-1 and claudin-5 in eyecups of rats (Figure 2i), indicating that BRB integrity was intact.

Several signaling pathways, including NF-κB (p65), PI3K/Akt, and mitogen-activated protein kinases (MAPK), have been demonstrated to be involved in the regulation of ABC transporter expressions [22,28,29,30]. Expressions of total and phosphorylated p38, Akt, ERK, p65, and JNK in rat eyecups were measured. It was found that BDL significantly increased phosphorylation of p38 and ERK but attenuated phosphorylation of JNK. However, levels of pp65 and pAkt were little altered by BDL (Figure 2j). MAPK-activated protein kinase 2 (MK2) is directly associated downstream to pp38 [31]. Here, phosphorylation of MK2 was also monitored to reflect activation of p38 MAPK. As expected, significantly increased levels of pMK2 (Figure 2k) were detected in eyecups of BDL rats. It was also noticed that BDL rats showed obviously lower protein expression of Brn-3a, a retinal ganglion cell (RGC) marker, inferring that BDL may lead to loss of RGC (Figure 2l).

### 2.3. Effects of Abnormally Altered Components in BDL Rat Serum on Function and Expression of MRP1 in ARPE-19 Cells

ARPE-19 cells serve as an in vitro outer BRB model to investigate the underlying mechanism by which BDL impairs the function and expression of retina MRP1. qRT-PCR analysis showed that, in line with findings in rat eyecups, high mRNA expression of *ABCC1* was detected in ARPE-19 cells (Figure 3a). The functionality of MRP1 was characterized by the intracellular accumulation of fluorescein and calcein with or without MK571 (50 μM), a widely used MRP inhibitor. Treatment with MK571 resulted in a 1.6-fold increase in intracellular fluorescein accumulation and a 2.1-fold increase in calcein accumulation in ARPE-19 cells, respectively (Figure 3b,c), confirming the functionality of MRP1. Coincubation with MK571 also obviously elevated intracellular UCB accumulation (Figure 3d), signifying that MRP1 contributes to UCB efflux in ARPE-19 cells. Next, the effect of serum from BDL rats on protein levels of MRP1 was evaluated. The results showed that incubation with medium containing 10% serum from BDL rats significantly decreased MRP1 protein expression (Figure 3e), implying that the abnormal components in BDL rat serum impaired MRP1 protein expression.

BDL rats often exhibit significant elevations in serum levels of UCB, ammonia, ADMA, and bile acids, which were also repeated in the study (Table 1). To investigate which altered components in BDL rat serum are factors impairing expression of MRP1, ARPE-19 cells were separately incubated with medium containing UCB, NH_4_Cl, ADMA, and bile acid cocktail for 72 h. The results showed that among the tested factors, only UCB concentration-dependently downregulated MRP1 protein expression in ARPE-19 cells (Figure 3f–i).

ARPE-19 cells were seeded in the transwell insert to form polarized RPE cell monolayers, which are structurally and functionally similar to the in vivo outer BRB [32]. Transepithelial electrical resistance (TEER) measurements and FITC-dextran permeability assay were conducted to investigate the barrier function of ARPE-19 cell monolayers. It was found that TEER reached a plateau on Day 6 and stayed stable from Day 6 to 10 (Figure 3j). Thus at 6 days after seeding, the monolayers with TEER values exceeding 90 Ω×cm^2^ were used for the following experiments. The permeability of FITC-dextran, the marker for the predominantly paracellular pathway transport, was measured. The P_app, A-to-B_ of FITC-dextran across the control insert membranes was 1.00 × 10^−5^ cm/s. However, in the presence of the ARPE-19 monolayers, the P_app, A-to-B_ of FITC-dextran was remarkably decreased to 1.18 × 10^−6^ cm/s (Figure 3k). These results revealed that ARPE-19 monolayers display the barrier function to substance transport.

It is generally accepted that MRP1 is mainly located at the basolateral membrane of RPE. Bidirectional transepithelial transport experiments with fluorescein were performed to confirm the MRP1 functionality of ARPE-19 cell monolayers. The results showed that the transport of fluorescein across ARPE-19 monolayers was asymmetric. P_app, A-to-B_ values (7.00 ± 1.94 × 10^−6^ cm/s) were significantly higher than P_app, B-to-A_ values (4.01 ± 0.82 × 10^−6^ cm/s), whose ratio (P_app, A-to-B_ /P_app, B-to-A_) was 1.75 (Figure 3l). Furthermore, MK571 inhibited the transport of fluorescein in the A-to-B direction, not the B-to-A direction (Figure 3l), indicating that MRP1 is involved in the A-to-B transport of fluorescein in ARPE-19 cells. The location of MRP1 protein on the ARPE-19 monolayer was detected using a confocal laser scanning microscope. A cross-section of z-stack images was reconstructed. The cross-sectional view of the x-z plane confirmed that MRP1 was mainly localized on the basolateral membrane of ARPE-19 cells (Figure 3m). Moreover, 72 h incubation with UCB also concentration-dependently decreased the transport of fluorescein in the A-to-B direction (Figure 3n). Consistently, immunofluorescence staining demonstrated that UCB significantly downregulated MRP1 protein levels (Figure 3o,p).

### 2.4. Involvement of P38 MAPK Pathway in UCB-Induced Downregulation of MRP1 Expression in APRE-19 Cells

In vivo data showed that BDL significantly increased phosphorylation of ERK, p38, and MK2 but decreased phosphorylation of JNK, inferring that these signal pathways could be involved in UCB-induced impairment of MRP1 expression. In APRE-19 cells, incubations with UCB (2, 10, 50 µM) (Figure 4a) or 10% BDL serum (Figure 4b) remarkably increased the phosphorylation of p38, MK2, Akt, and ERK1/2. P38 inhibitors SB203580 or SB202190 but neither ERK1/2 inhibitor U0126 nor Akt inhibitor LY294002 reversed the downregulation of MRP1 protein expression by UCB (Figure 4c,d). UCB itself increased p38 phosphorylation (Figure 4e). Neither SB203580 nor SB202190 reversed the phosphorylation of p38 by UCB. In contrast, UCB and SB202190 showed synthetic effects on p38 phosphorylation (Figure 4e). UCB markedly increased MK2 phosphorylation, which was almost abolished by SB203580 or SB202190 (Figure 4f). This finding is in agreement with the previous studies that SB203580 and SB202190 inhibit p38 MAPK catalytic activity, not phosphorylation of p38 [33,34]. To further verify whether p38 MAPK is involved in the impairment of MRP1 expression, the effect of anisomycin, a well-known p38 activator, on MRP1 protein expression was documented. This was consistent with findings in treatment with UCB that neither SB202190 nor SB203580 reversed anisomycin-induced phosphorylation of p38 but almost abrogated anisomycin-induced upregulation of pMK2 levels and impairment of MRP1 expression (Figure 4g).

The role of p38 MAPK in UCB-induced MRP1 downregulation was further confirmed using p38 knockdown with siRNA. Transfection of p38 siRNA markedly attenuated p38 protein expression, which was 45% of cells transfected with negative control siRNA, demonstrating successful silencing of p38 (Figure 4h). Both UCB and anisomycin no longer decreased MRP1 protein levels in cells transfected with p38 siRNA (Figure 4i).

Roles of reactive oxygen species (ROS) in UCB-induced MRP1 downregulation were also documented. APRE-19 cells were incubated with medium containing UCB, and H_2_O_2_ (0.1 mM) served as positive control. ROS levels were determined using fluorescent probe DCFH-DA according to the manufacturer’s instruction (Beyotime, Shanghai, China) with excitation/emission wavelengths of 480/535 nm on a SpectraMax Gemini XPS microplate fluorometer. The results showed that UCB at the tested concentrations (2~50 μM) did not prevent H_2_O_2_-induced ROS formation and did not show any antioxidant effects (Figure 4j). On the contrary, 50 μM UCB enhanced formation of ROS, which was attenuated by N-acetyl-L-cysteine (NAC, 8 mM) (Figure 4k). UCB-induced downregulation of MRP1 protein was not attenuated by NAC (Figure 4l). Furthermore, H_2_O_2_ did not affect expressions of MRP1 protein, indicating that downregulation of MRP1 protein by UCB was independent of ROS.

### 2.5. Effect of Bilirubin on the Function and Expression of Mrp1 at Rat BRB

The effect of intraperitoneal (i.p.) injection of bilirubin on the expression and function of Mrp1 in the retina of rats was investigated. Treatment with bilirubin significantly increased levels of UCB in rat plasma (Figure 5a), indicating successful development of HB rats. HB rats also showed markedly increased total bilirubin, liver weight/body weight ratio, and spleen weight/body weight ratio, but levels of ALT, AST, and AKP were unaltered (Table 2). This was similar to findings in BDL rats that bilirubin treatment impaired function and expression of Mrp1 in rat retina. K_r/p_ of fluorescein and DNP-SG were increased to about 150.9% and 161.8% of control rats (Figure 5d,g), respectively. The levels of UCB were also increased from less than LLOQ (0.67 pmol/mg protein) in the retina of control rats to 2.55 ± 1.07 pmol/mg protein in HB rats (Figure 5h). Consistently, expressions of retinal Mrp1 were remarkably decreased to 56.6% of control rats (Figure 5i). Bilirubin treatment also increased phosphorylation of p38 and MK2 in rat eyecups (Figure 5i) and significantly reduced protein expression of Brn-3a (Figure 5j).

## 3. Discussion

The main finding of the study was that BDL significantly downregulated the function and expression of Mrp1 at rat BRB. Immunofluorescence analysis showed that Mrp1 was mainly colocated with Glut1 at the basolateral membrane of rat RPE, where it mediates efflux of its substrates from the retina via the outer BRB [14].

Retina distributions of both fluorescein and DNP-SG, substrates of Mrp1, were measured to assess the function of retina Mrp1. Although Mrp2 also transports fluorescein and DNP-SG [14], levels of retina *Abcc2* mRNA were lower one-thousandth times than that of retina *Abcc1* mRNA in rats, indicating that transports of fluorescein and DNP-SG across BRB of rats are mainly mediated by Mrp1. Immunofluorescence staining demonstrated colocation of Mrp1 protein with Glut1 protein at the basolateral side of rat RPE, which is consistent with the location of Mrp1 in mouse retina reported previously [24]. In line with decreases in expression of retina Mrp1, BDL significantly increased K_r/p_ of fluorescein and DNP-SG. BDL did not affect the permeability of FITC-dextran across BRB nor expressions of tight junction proteins ZO-1 and claudin-5, indicating that BRB integrity of BDL rats remains intact and that the increased retina penetrations of fluorescein and DNP-SG in BDL rats are mainly attributed to decreases in expressions of Mrp1 protein.

It was also noticed that BDL little affected plasma levels of fluorescein but significantly lowered plasma levels of DNP-SG in rats. Fluorescein was mainly eliminated via urine [35] and BDL little affected renal function, which may explain that BDL did not affect levels of plasma fluorescein [36]. Formation of DNP-SG from 1-chloro-2,4-dinitrobenzene is mediated by glutathione-S-transferases (GSTs). DNP-SG is further metabolized by γ-glutamyltransferase. Several reports have demonstrated that animals [37,38,39,40,41] and humans [42] suffering from liver injury show decreases in hepatic output and plasma concentration of GSH and the increases in activity of γ-glutamyltransferase, which contributes to the lower plasma levels of DNP-SG in BDL rats and is also supported by the report [41].

Mrp1 is predominantly expressed in RPE. ARPE-19 cell line, a commonly used in vitro model for the study of RPE, shows similar expression patterns of ABC transporters with primary human RPE cells [13]. Here, ARPE-19 cells were used to investigate factors that impair expression of MRP1 protein. High expression of *ABCC1* mRNA was detected in ARPE-19 cells, and MRP inhibitor MK571 also increased intracellular accumulation of fluorescein and calcein, confirming the expression and function of MRP1 in ARPE-19 cells. As expected, incubation with serum from BDL rats markedly decreased MRP1 protein expression, inferring the existence of MRP1 modulators in serum from BDL rats. Effects of the increased components (such as UCB, ammonia, ADMA, and bile acids) in serum from BDL rats on MRP1 protein expressions were further investigated. Only UCB significantly suppressed MRP1 protein expression in a dose-dependent manner, indicating that the increased levels of UCB are the main factor suppressing expression of retinal Mrp1 in BDL rats. Significantly increased levels of UCB were also observed in plasma of BDL rats, near to the tested levels of UCB in vitro cells, further confirming the contribution of UCB to downregulation of retinal Mrp1. However, reports about the effects of UCB on MRP1 expression are often confusing [43,44,45,46]. This was consistent with our results that UCB downregulated protein expression of Mrp1 in rat choroid plexus epithelial cells [43]. But in rat liver and spleen, bilirubin upregulated mRNA and protein expressions of Mrp1 [44]. In cultured mouse astroglial cells, bilirubin enhanced expression of Mrp1 on the plasma membrane [45]. In primary cultures of rat neurons or astrocytes, bilirubin did not affect Mrp1 expression [46]. These discrepancies may come from species or tissue specificity.

Further study showed that UCB markedly increased phosphorylation of ERK1/2 and p38 in APRE-19 cells, which were in line with in vivo data, but only p38 inhibitors can restore UCB-induced downregulation of MRP1 and increases in MK2 phosphorylation. Roles of p38 in UCB-induced downregulation of MRP1 were further confirmed using both p38 activator and p38 silencing. p38 activator anisomycin also downregulated MRP1 protein expression, accompanied by the increased expression of pMK2. p38 silencing almost abolished the downregulation of MRP1 by both bilirubin and anisomycin. Bilirubin-treated rats were developed further to confirm the roles of UCB in the regulation of retina Mrp1. This was consistent with in vitro data that treatment with bilirubin significantly downregulated expression and function of Mrp1 in rat retina, accompanied by increases in phosphorylation of p38 and MK2. All these results gave the conclusion that BDL significantly downregulated expression of MRP1 by activating the p38 pathway by the elevated UCB.

UCB, behaving like a double-edged sword, shows its cytotoxic and cytoprotective effect [47] by affecting intracellular ROS formation. The present study also showed that UCB may serve as a ROS generator to increase ROS formation. Several reports have demonstrated that UCB may increase ROS formation [48,49,50,51] or reduce ROS levels [48,51,52], which are dependent on UCB levels or cell status (/or targeted tissues or cell types). For example, heart endothelial (H5V), kidney tubular (HK2), and neuronal (SH-SY5Y) cell lines and high concentrations of UCB (>3.6 µM in SH-SY5Y cells, >15 µM in H5V cell, and >15 µM in HK2 cells) significantly increased intracellular ROS production, but this phenomenon was not observed in HepG2. On the contrary, low concentrations of UCB showed antioxidant effects in the four cells [51]. Effects of UCB on intracellular ROS levels in bEnd3 (microvascular endothelial cell line from mouse brain) were dependent on glucose levels. UCB little affected ROS production in bEnd3 treated with a low glucose level (5.5 mM). In bEnd3 treated with high glucose level (25 mM), low “physiological” concentrations of UCB (0.1~2 µM) attenuated high glucose-induced ROS production, while higher UCB concentrations markedly elevated ROS production. In MS1 cells (microvascular endothelial cell line from mouse pancreatic islets), low concentrations of UCB (0.1~5 µM) reversed the increases in ROS by high glucose (25 mM), while further increases in UCB concentration did not cause further changes in ROS production in the MS1 cells [48]. Several reports have demonstrated that bilirubin concentration has been negatively associated with the risk of diabetic retinopathy or diabetic neuropathy [53,54,55], and baseline total bilirubin levels had a U-shaped relationship with incidence of diabetic retinopathy, whose point of inflection was 12.60~13.80 µM [55]. BDL rats showed significantly serum levels of total bilirubin (about 91.50 µM), indicating that under liver failure, bilirubin, mainly serving as a ROS generator, induced ROS formation, in turn, impairing the retina.

Roles of ROS in bilirubin-induced downregulations of MRP1 were also documented. Bilirubin induced ROS formation and impaired expression of MRP1, accompanied by increases in expression of pMK2. Although NAC attenuated the increases in ROS by bilirubin, NAC did not obviously reverse expression of MRP1. H_2_O_2_ itself did not affect expression of MRP1. All of these indicated that bilirubin downregulated expression of MRP1 independent of the ROS pathway.

Both BDL and treatment with UCB significantly increased levels of UCB in the retina and plasma of rats. Bilirubin, a known toxic product of heme catabolism, is an endogenous substrate of MRP1/Mrp1 [45,46,56,57], indicating that the increased retinal UCB levels in BDL rats and HB rats are attributed to downregulation of retinal Mrp1 protein and increased plasma UCB levels. Roles of MRP1/Mrp1 expression in susceptibility of multiple neuronal cell types to unconjugated bilirubin have been demonstrated, including primary cultures of rat neurons [46], astrocytes [46], and human SH-SY5Y cells [57]. An in vitro study showed that exposure of murine retina to bilirubin impaired neurotransmission and reduced b-wave amplitude [58]. The present study also showed that in line with the evaluated levels of retinal bilirubin, both BDL and bilirubin treatment remarkably downregulated expression of Brn-3a protein, inferring loss of ganglion. Thus, the roles of the elevated levels of retinal UCB in hepatic retinopathy need further investigation.

In summary, BDL significantly downregulated expression of retina Mrp1 in rats, which was partly attributed to the activation of p38 MAPK by the elevated bilirubin.

## 4. Methods and Materials

### 4.1. Reagents

DNP-SG was supplied by Toronto Research Chemicals (Toronto, ON, Canada). 1-Chloro-2,4-dinitrobenzene and FITC-dextran (3–5 kDa) were purchased from Sigma Aldrich (St. Louis, MO, USA). ADMA, SDMA, U0126, LY294002, anisomycin, and MK571 were obtained from EFEBIO (Shanghai, China). Bilirubin, fluorescein sodium, sodium cholate, sodium chenodeoxycholate, sodium deoxycholate, sodium hyodeoxycholate, SB202190, and SB203580 were from Aladdin (Shanghai, China). Calcein-AM was acquired from Beyotime (Shanghai, China). All other reagents were of analytical grade and available from commercial sources.

### 4.2. Animals

Male Sprague–Dawley rats, weighing 180 g to 200 g, were from Sino-British Sippr/BK Laboratory Animal Ltd. (Shanghai, China). The rats were housed in a room with controlled temperature (24 ± 2 °C) and humidity (50 ± 5%) under a 12 h light/dark cycle. Animals were fed with a commercial chow diet and water ad libitum. The animal studies were performed in accordance with the Guide for the Care and Use of Laboratory Animals (National Institutes of Health) and were approved by the Animal Ethics Committee of China Pharmaceutical University (Approval Number: 202004010).

### 4.3. Development of BDL Rats

Development of BDL rats was performed as described previously [18]. Briefly, rats were anesthetized using an intraperitoneal dose of pentobarbital (40 mg/kg), and then the common bile duct was exposed and ligated with 4-0 silk. Sham rats underwent a similar procedure without ligation. Following closing abdominal incision, the rats were kept on warm pads (37 °C) to recover. Two weeks following surgery, BDL and Sham rats were selected for the following experiments.

### 4.4. Distributions of Fluorescein and DNP-SG in Rat Retina

Distributions of fluorescein and DNP-SG, two typical substrates of Mrp1 [17,59,60], were utilized to assess the function of Mrp1 at the BRB of rats. Briefly, experimental rats were sacrificed under light ether anesthesia at 30 min following i.v. administration of 2 mg/kg fluorescein or at 15 min following i.v. administration of 5 mg/kg 1-chloro-2,4-dinitrobenzene (a precursor of DNP-SG). Blood samples and tissue (liver, spleen, and eye) were collected. Plasma and serum samples were obtained to analyze substrates, ADMA, SDMA, and serum biochemical parameters. Serum biochemical parameters such as ALT, AST, ALP, total bilirubin, ammonia, and total bile acids were measured using commercially available kits (Jiancheng Bioengineering Institute, Nanjing, China) according to the manufacturer’s instructions. Eyecups (RPE/choroid/sclera and neural retina) were prepared by removing the cornea, lens, and vitreous from enucleated eyes [61]. For substrate analysis, the neural retina was further dissected from the eyecup.

Another batch of rats was used to assess BRB permeability using FITC-dextran (3–5 kDa) as a probe, according to the previous report [62]. In brief, the experimental rats were sacrificed under light ether anesthesia at 30 min following i.v. FITC-dextran (50 mg/kg). Fluorescence was determined (excitation 485 nm, emission 525 nm) with a SpectraMax Gemini XPS microplate fluorometer (Sunnyvale, CA, USA), and the amount of FITC-dextran in retina and plasma was calculated based on the corresponding standard curve.

### 4.5. Drug Assays

Plasma ADMA and SDMA were measured by HPLC with *ortho*-phthalaldehyde derivatization [63]. Concentrations of fluorescein in plasma and retina were measured by HPLC-MS following the previously established method [28]. Concentrations of UCB in plasma and retina were measured using the HPLC-UV method [64]. Retina UCB values were normalized to protein content and expressed as picomole per milligram protein.

The HPLC-MS/MS method was developed to measure concentrations of DNP-SG in plasma and retina. For the retina, the neural retina was homogenized on ice in 300 μL 50% acetonitrile, and 100 μL of homogenates were mixed with 50 μL acetonitrile containing 25 ng/mL internal standard (4-hydroxydiclodenac). For plasma, 50 μL of plasma was mixed with 100 μL acetonitrile containing 500 ng/mL internal standard. The above mixture was centrifuged at 15,000× *g* for 10 min at 4 °C. Two μL of the supernatant was injected into the HPLC-MS/MS system. Analysis was performed by negative ion LC-MS/MS electrospray (ABSciex QTrap 4500) on a Shim-pack VP-ODS column (5 µm, 150 L × 2.0 mm, Shimadzu, Japan). For the gradient elution, buffer A was 0.001% formic acid in the water, and buffer B was acetonitrile. The flow rate was 0.3 mL/min. A gradient program started at 20% B and changed linearly to 70% B at 1.5 min, then linearly to 90% B at 4 min, where it was run isocratically until 4.2 min. From 4.2 min to 4.3 min, the gradient was linearly switched back to starting conditions, keeping isocratically until 6.2 min. Calibration ranges were from 62.5 to 8000 ng/mL for plasma and from 0.98 to 125 ng/mL for retina homogenate (equivalent to 0.195~25 ng/mg protein), respectively.

### 4.6. Immunofluorescence Staining

Paraffin-embedded sections (3 μm) of rat eyecups were deparaffinized in dimethyl benzene and rehydrated in decreasing ethanol. The sections were blocked with 5% goat serum and incubated with mixed primary antibodies of mouse anti-Mrp1 (1:100, Abcam, Cambridge, UK) and rabbit anti-Glut1 (1:400, Servicebio) overnight at 4 °C. Sections, following washing with phosphate-buffered saline (PBS) three times, were incubated for 1 h at room temperature with Alexa Fluor 488-conjugated goat anti-mouse IgG (1:1000, Invitrogen, Waltham, MA, USA) and Alexa Fluor 594-conjugated goat anti-rabbit IgG (1:1000, Invitrogen). The cell nuclei were counterstained with 4’,6-diamidino-2-phenylindole (DAPI, Servicebio). Images were captured on LSM700 confocal laser scanning microscope (Zeiss). Integrated densities for Mrp1 in the RPE were quantified using ImageJ software (NIH, Bethesda, MD, USA) and normalized by RPE length.

ARPE-19 cells grown on the filter were fixed with 4% formaldehyde for 15 min at room temperature. Then, cells with the filter were cut out and stained with anti-MRP1 primary antibody (1:100, CST) at 4 °C overnight, followed by secondary antibody and DAPI. Finally, cells were mounted on the slide and viewed under confocal microscopy. The fluorescence intensity of MRP1/DAPI was quantified by ImageJ.

### 4.7. Quantitative Real-Time Polymerase Chain Reaction (qRT-PCR)

Total RNA was isolated from eyecups of normal rats and ARPE-19 cells (Cellcook, Guangzhou, China) using RNAiso Plus Reagent and reverse-transcribed into cDNA using HiScript^®^ III RT SuperMix (Vazyme, Nanjing, China). Expressions of mRNA, including *Abcb1a/1b*/*ABCB1*, *Abcg2*/*ABCG2*, and *Abcc1~6*/*ABCC1-6*, were determined by qRT-PCR. *Actb*/*ACTB* was used as a housekeeping gene to normalize the detected gene. qRT-PCR was conducted with SYBR Master Mix (Yeasen) on an Applied Biosystems QuantStudio 3 Real-Time PCR System (Thermo Fisher Scientific). Analyses used the 2^−ΔCt^ method [65], where ΔCt = Ct (*target gene*) − Ct (*Actb*). Expression levels (2^−ΔCt^) of these genes were plotted on a logarithmic scale. The sequences of rat and human primers are presented in Table 3.

### 4.8. Cell Culture and Drug Treatment

ARPE-19 cells were cultured in Dulbecco’s Modified Eagle Medium (DMEM)/F12 medium (Invitrogen) supplemented with 10% fetal bovine serum (FBS) (Invitrogen), 100 IU/mL penicillin and 100 μg/mL streptomycin under 5% CO_2_ in an incubator (37 °C). The cells were seeded in 12-well plates at a density of 1 × 10^5^ cells/well. At 70% confluence, ARPE-19 cells were exposed to culture medium containing 10% serum from rats or the tested factors, including NH_4_Cl (0.2, 1, and 5 mM), UCB (2, 10, and 50 μM), ADMA (2, 10, and 50 μM), and bile acid cocktail (50 and 100 μM), respectively. Levels of NH_4_Cl and UCB were referenced from the literature [18,22]. Levels of ADMA and bile acid cocktail were designed according to the measured levels in BDL rats. Bile acid cocktail consisted of sodium cholate, sodium chenodeoxycholate, sodium deoxycholate, and sodium hyodeoxycholate (30:18:8:5), which was obtained from a previous report [66]. CCK 8 test showed that these agents at tested concentrations used in the study had no damage to the viability of cultured cells. Following 72 h incubation, the cells were used to assess the expression and function of MRP1 protein.

### 4.9. P38 Knockdown with siRNA

ARPE-19 cells were transfected with p38 siRNA (90 pmol per 12-well) using lipofectamine 3000 (Invitrogen) according to instructions. The sequence of p38 siRNA was 5′- GCAUAAUGGCCGAGCUGUUTT -3′, which was referenced from the literature [67]. Both siRNA targeting p38 and scramble siRNA control were synthesized by GenePharma (Shanghai, China). After being transfected with siRNA for 24 h, cells were exposed to UCB (50 μM) or anisomycin (50 nM) for another 72 h. Then, Western blot was performed to detect the p38 and MRP1 expressions.

### 4.10. Uptake of MRP1 Substrates by ARPE-19 Cells

MRP1 functionality of ARPE-19 cells was assessed using accumulation of fluorescein and calcein. Briefly, cells were washed with Hank’s balanced salt solution (HBSS) 2 times and incubated with HBSS containing fluorescein (50 μM) or calcein-AM (0.5 μM) in the presence or absence of MRP1 inhibitor MK571 (50 μM). Following 30 min incubation, uptake reaction was stopped via aspirating solution, followed by washing thrice with ice-cold HBSS. Fluorescence intensity of intracellular calcein (excitation 494 nm; emission 514 nm) was measured by SpectraMax Gemini XPS microplate fluorometer (Sunnyvale, CA, USA) and normalized by cellular protein content. Intracellular fluorescein content was determined by HPLC-MS [28] and also normalized by cellular protein content. Protein contents were determined by the Coomassie protein assay reagent (Beyotime).

Whether MRP1 mediated the transport of UCB in ARPE-19 cells was also investigated. Cells were incubated with HBSS containing UCB (4 and 10 μM) for 10 min with or without MK571 (50 μM) [68], and cellular amount of UCB was measured using the HPLC-UV method [64].

### 4.11. Transport of Fluorescein across ARPE-19 Monolayer

ARPE-19 cells were seeded at 1 × 10^5^ cells/well on transwell inserts (PET membrane, 0.4 μm pore, SPL 37024, Korea). Cell monolayer integrity and paracellular permeability were assessed by TEER and permeability of FITC-dextran, respectively [69]. TEER was measured using a Millicell ERS-2 Voltohmmeter (Merck, Kenilworth, NJ, USA). The measured resistance value was multiplied by the surface area of the membrane (0.33 cm^2^) to obtain TEER (Ω × cm^2^). Monolayer with TEER over 90 Ω × cm^2^ was used for further experiments. For FITC-dextran permeability assays, 0.35 mL of FITC-dextran (1 mg/mL) was added to the apical compartment, and 1.3 mL of HBSS solution was added to the basolateral compartment. The samples in the lower chambers were taken after 30 min incubation and the amount of FITC-dextran was determined by the SpectraMax Gemini XPS microplate fluorometer (excitation 480 nm, emission 520 nm). Apparent permeability coefficients (P_app_) were calculated as P_app_ = (A_receptor_/1800)/(Area × C_0,donor_) [70], where A_receptor_/1800, Area, and C_0,donor_ are flux rate (μg/s), the surface area of membrane (0.33 cm^2^), and the initial concentration of substrate in donor chamber (μg/mL), respectively.

Bidirectional transport of fluorescein across the ARPE-19 monolayer was carried out. Fluorescein (50 μM) was added to the apical compartment or basolateral compartment in the absence or presence of MK571 (50 μM). Concentrations of fluorescein in the receiver were measured following 30 min incubation. Both P_app_, _A-to-B_ and P_app_, _B-to-A_ values were estimated.

### 4.12. Western Blot

Protein expressions in eyecups or cells were measured using Western blot. Total protein was extracted using the Blue Loading Buffer Pack (CST) containing protease inhibitor cocktail (Yeasen) and phenylmethyl sulfonyl fluoride. Proteins were separated on 8~12% SDS-PAGE and transferred to a nitrocellulose membrane (Pall Life Science, New York, NY, USA). Following blocking for 2 h at room temperature with 5% non-fat dry milk, the blots were incubated overnight at 4 °C with primary antibodies: anti-Mrp1 (1:500, Abcam), anti-MRP1 (1:1000, CST), anti-ERK1/2 (1:1000, CST), anti-phospho(p)ERK1/2 (1:1000, CST), anti-p38 MAPK (1:1000, CST), anti-pp38 MAPK (1: 1000, CST), anti- NF-κB p65 (1:1000, CST), anti-pp65 (1:1000, CST), anti-Akt (1: 1000, CST), anti-pAkt (1:1000, CST), anti-JNK1/2/3 (1:500, Huaan Biotechnology), anti-p-JNK1/2/3 (1: 500, Huaan Biotechnology), anti-MK2 (1:500, CST), anti-pMK2 (1: 500, CST), anti-ZO-1 (1:1000, Wanleibio), anti-claudin-5 (1:1000, Wanleibio), anti-Brn-3a (1:200, Santa), and β-actin (1:8000, Proteintech). After washing with TBST, blots were incubated with HRP-conjugated anti-rabbit IgG antibody (1:3000, CST) or HRP-conjugated anti-mouse IgG antibody (1:3000, CST) for 1.5 h at room temperature. Finally, protein amount was detected by the ECL reagent (Vazyme Biotech, Nanjing, China). Densitometric analysis was performed using Image J. Protein levels were normalized to either total protein or β-actin levels.

### 4.13. Development of HB Rats

HB rats were developed according to the previous report [18]. Briefly, 12 rats were randomly divided into bilirubin-treated and control groups. Bilirubin-treated rats received i.p. bilirubin (85.5 μmol/kg/d) injections consecutively for 14 days. Control rats only received vehicle. At 24 h following last dose, expression and function of Mrp1 in the retina of rats were assessed as described above.

### 4.14. Statistical Analysis

All data are expressed as the mean ± S.D. (standard deviation). Statistical analyses were conducted using GraphPad Prism 8.0.2 software by unpaired 2-tailed Student’s *t*-test or 1-way ANOVA followed by the Tukey test. *p* < 0.05 was considered statistically significant.

## Figures and Tables

**Figure 1 ijms-23-07666-f001:**
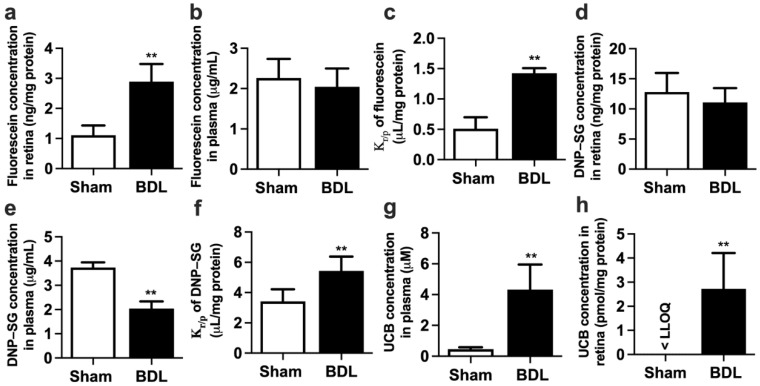
Effect of BDL on the function of Mrp1 in rat retina. (**a**) Retina fluorescein levels, (**b**) plasma fluorescein levels, and (**c**) retina-to-plasma ratio (K_r/p_) of fluorescein at 30 min following intravenous (i.v.) fluorescein (2 mg/kg) to rats. (**d**) Retina DNP-SG levels, (**e**) plasma DNP-SG levels, and (**f**) K_r/p_ of DNP-SG at 15 min following i.v. 1-chloro-2,4-dinitrobenzene (5 mg/kg) to rats. (**g**) Plasma UCB levels and (**h**) retina UCB levels in Sham and BDL rats. Lower limit of quantitation (LLOQ) = 0.67 pmol/mg protein. Data are expressed as means ± S.D. (*n* = 6). ******
*p* < 0.01 vs. Sham rats.

**Figure 2 ijms-23-07666-f002:**
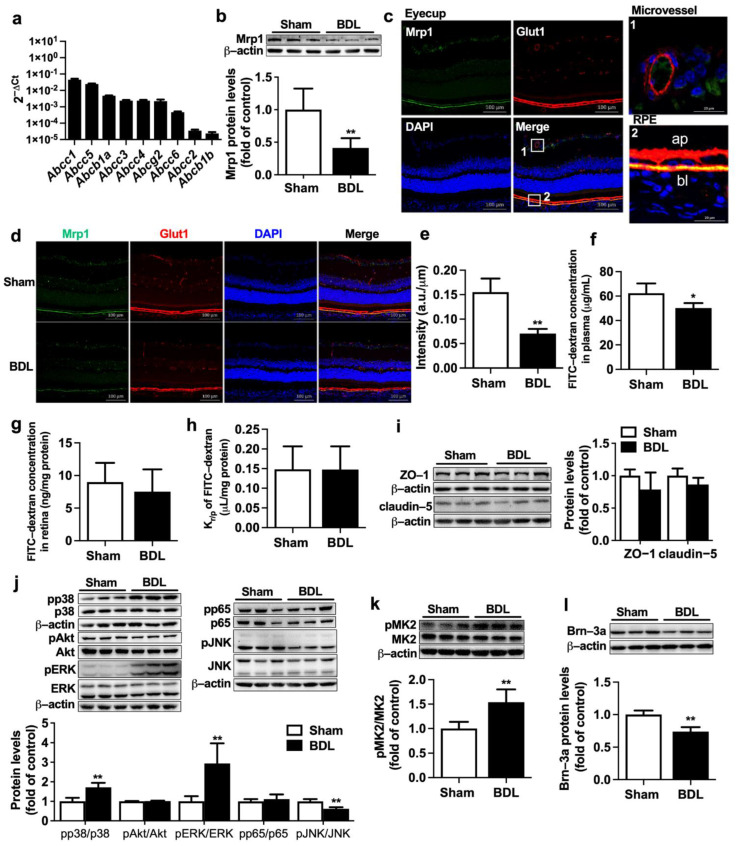
Effect of BDL on expression of Mrp1 in rat retina. (**a**) The mRNA levels of *Abcc1~6*, *Abcb1a/1b*, and *Abcg2* in eyecups of normal rats were measured by qRT-PCR and normalized to *Actb* (2^−ΔCt^). (**b**) Expression of Mrp1 protein in eyecups of Sham and BDL rats (*n* = 6). (**c**) The location of Mrp1 (green) and Glut1 (red) in the eyecup of rats (Scale bar = 100 μm). Nuclei are DAPI-stained (blue). Regions marked 1 and 2 indicate microvessel and RPE, respectively, which are detailed in the upper right and lower right (Scale bar = 20 μm). Ap, apical membrane; bl, basolateral membrane. (**d**) Immunostaining analysis of Mrp1 and Glut1 in eyecups (Scale bar = 100 μm) and (**e**) its integrated fluorescence intensity of Mrp1 normalized by RPE length (*n* = 3). (**f**) Plasma FITC-dextran levels, (**g**) retina FITC-dextran levels, and (**h**) K_r/p_ of FITC-dextran in Sham and BDL rats at 30 min following i.v. 50 mg/kg FITC-dextran (*n* = 5). (**i**) Expressions of ZO-1 and claudin-5 (*n* = 6). (**j**) Immunoblots for pp38, p38, pAkt, Akt, pERK, ERK, pp65, p65, pJNK, and JNK and ratios of phosphorylated proteins to total protein levels (*n* = 6). (**k**) Expressions of pMK2 and total MK2 levels (*n* = 6). (**l**) Expression of Brn-3a protein (*n* = 6). Data are expressed as the mean ± S.D. *****
*p* < 0.05, ******
*p* < 0.01 vs. Sham rats.

**Figure 3 ijms-23-07666-f003:**
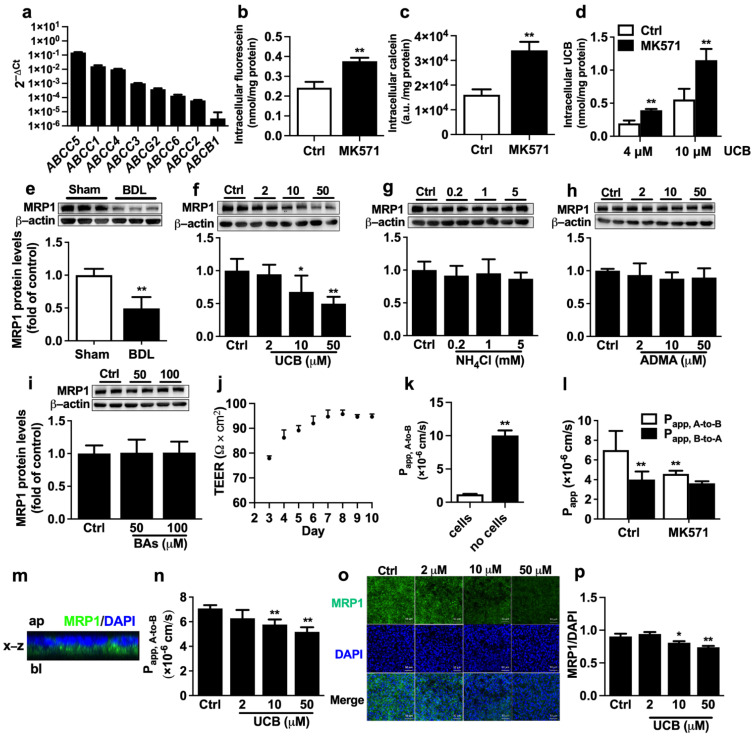
Effect of UCB on the function and expression of MRP1 in ARPE-19 cells. (**a**) The mRNA levels of efflux transporters in ARPE-19 cells were normalized to the *ACTB* (2^−ΔCt^) (*n* = 3). Effects of MRP1 inhibitor MK571 on cellular uptakes of (**b**) fluorescein (*n* = 6), (**c**) calcein (*n* = 6), and (**d**) UCB (*n* = 4) in ARPE-19 cells. (**e**) Effect of serum from BDL rats on MRP1 protein expression (*n* = 6). Effects of (**f**) UCB, (**g**) NH_4_Cl, (**h**) ADMA, and (**i**) bile salt cocktail on MRP1 protein expressions (*n* = 6). (**j**) TEER values across ARPE-19 monolayer after seeding (*n* = 4). (**k**) Apical (A) -to-basolateral (B) transport of FITC-dextran across ARPE-19 monolayer and control insert membranes (*n* = 4). (**l**) A-to-B and B-to-A transport of fluorescein with or without MK571 (*n* = 6). (**m**) Cross-sectional view of the reconstructed z-stack images in x-z plane. Ap, apical; bl, basolateral. (**n**) A-to-B transport of fluorescein after pre-treatment with UCB for 72 h (*n* = 4). (**o**) Protein levels of MRP1 were accessed by immunofluorescence after incubation with UCB (Scale bar = 50 μm). (**p**) The mean MRP1 fluorescence intensity was normalized to the mean DAPI fluorescence intensity (*n* = 3). Data are expressed as the mean ± S.D. *****
*p* < 0.05, ******
*p* < 0.01 vs. Sham serum, control (Ctrl) cells or P_app, A-to-B_ of Ctrl.

**Figure 4 ijms-23-07666-f004:**
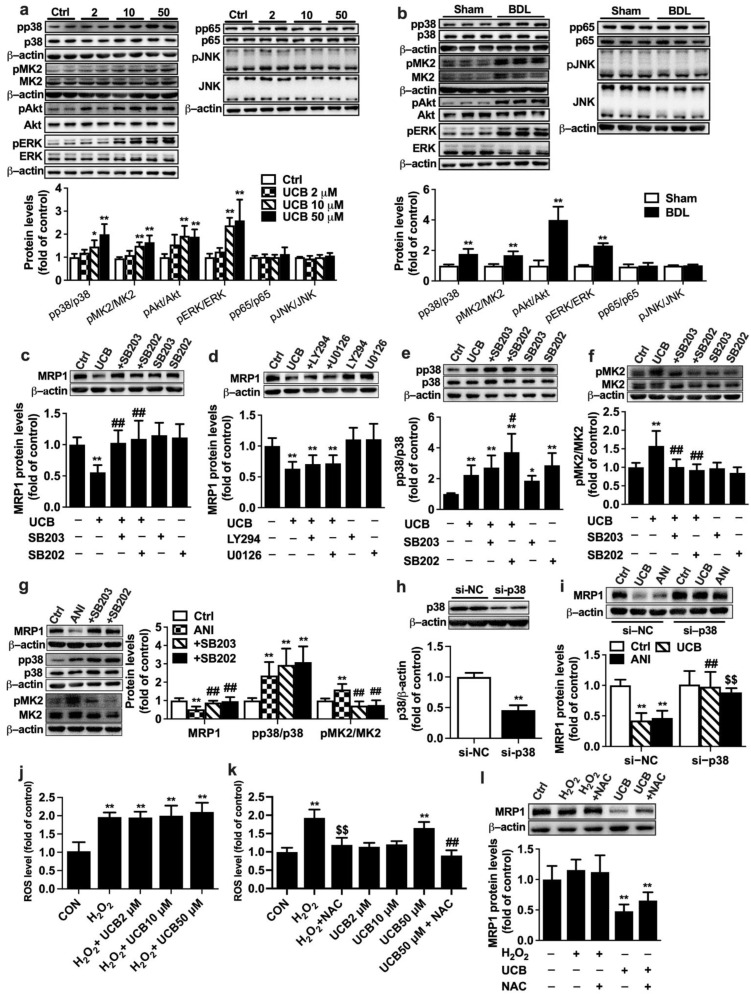
Roles of signal pathways in impairment of MRP1 protein expression by UCB in APRE-19 cells. Effects of (**a**) UCB and (**b**) BDL rat serum on expressions of pp38/p38, pMK2/MK2, pAkt/Akt, pERK/ERK1/2, pp65/p65, and pJNK/JNK levels. Effects of (**c**) p38 inhibitors SB203580 (SB203) and SB202190 (SB202), (**d**) Akt inhibitor LY294002 (LY294), and ERK1/2 inhibitor U0126 on UCB-induced downregulation of MRP1 protein. Effects of SB203 and SB202 on increases in phosphorylation of (**e**) p38 and (**f**) MK2 levels by UCB. (**g**) Effects of SB203 and SB202 on anisomycin (ANI)-induced impairment of MRP1 protein expression and increases in phosphorylation of p38, MK2. (**h**) Validation of p38 siRNA. (**i**) Effects of p38 siRNA on UCB- and ANI-induced decreases in MRP1 protein expressions. (**j**) Effects of UCB on H_2_O_2_ -induced ROS formation. (**k**) Effects of NAC on UCB-induced ROS formation. (**l**) Effect of H_2_O_2_ on MRP1 protein levels. Data are expressed as the mean ± S.D. (*n* = 6). * *p* < 0.05, ** *p* < 0.01 vs. control (Ctrl) cells, Sham serum, si-negative control (si-NC) + Ctrl. ^#^
*p* < 0.05, ^##^
*p*< 0.01 vs. UCB (50 μM), ANI or si-NC + UCB. ^$$^
*p* < 0.01 vs. si-NC + ANI or H_2_O_2_.

**Figure 5 ijms-23-07666-f005:**
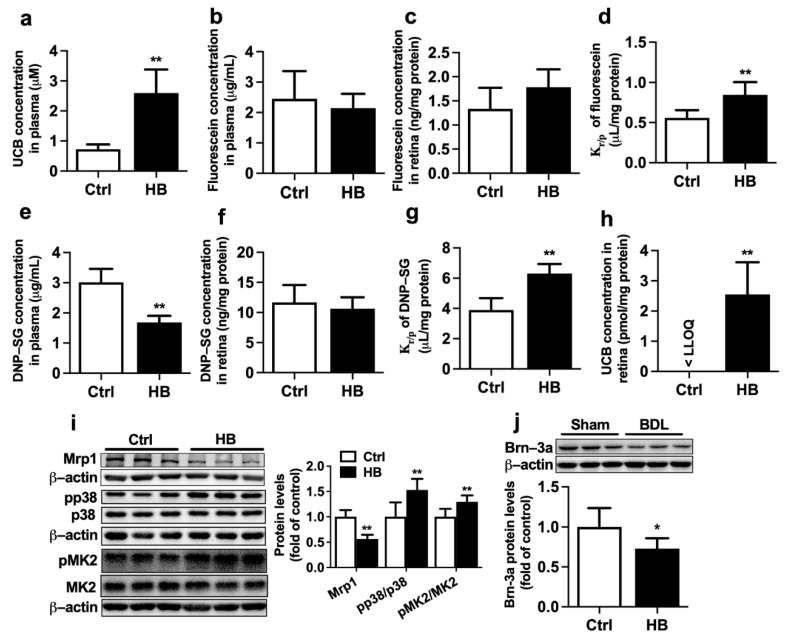
Effects of bilirubin (UCB) treatment on function and expression of Mrp1 in rat retina. (**a**) Levels of UCB in plasma of control (Ctrl) and hyperbilirubinemia (HB) rats. (**b**) Plasma fluorescein levels, (**c**) retina fluorescein levels, and (**d**) K_r/p_ of fluorescein at 30 min following i.v. fluorescein (2 mg/kg) to Ctrl and HB rats. (**e**) Plasma DNP-SG levels, (**f**) retina DNP-SG levels, and (**g**) K_r/p_ of DNP-SG at 15 min following i.v. 1-chloro-2,4-dinitrobenzene (5 mg/kg) to Ctrl and HB rats. (**h**) Retina UCB levels in Ctrl and HB rats. LLOQ = 0.67 pmol/mg protein. (**i**) Expressions of Mrp1, pp38/p38, and pMK2/MK2 levels in eyecups of Ctrl and HB rats. (**j**) Brn-3a protein expression in retina. Data are expressed as the mean ± S.D. (*n* = 6). *****
*p* < 0.05, ******
*p* < 0.01 vs. Ctrl.

**Table 1 ijms-23-07666-t001:** Physiological and biochemical parameters in Sham and BDL rats.

Parameters	Sham Rats	BDL Rats
Body weight (BW) (g)	225.0 ± 19.9	231.8 ± 15.5
Liver weight (%BW)	2.84 ± 0.15	6.84 ± 0.96 **
Spleen weight (%BW)	0.23 ± 0.04	0.55 ± 0.17 **
ALT (IU/L)	6.45 ± 4.55	22.47 ± 11.37 **
AST (IU/L)	20.22 ± 6.00	90.99 ± 31.51 **
ALP (IU/L)	114.90 ± 25.80	254.80 ± 47.67 **
Total bilirubin (µmol/L)	ND ^a^	91.50 ± 15.50 **
Total bile acids (µmol/L)	29.50 ± 7.48	133.84 ± 23.95 **
Serum ammonia (µmol/L)	210.41 ± 27.41	276.89 ± 45.35 *
ADMA (ng/mL)	174.49 ± 32.84	277.29 ± 52.74 **
SDMA (ng/mL)	25.14 ± 11.45	37.71 ± 16.28

^a^, ND: <3.4 µmol/L; ADMA, asymmetric dimethylarginine; SDMA, symmetric dimethylarginine. Data are expressed as mean ± S.D. (*n* = 6). * *p* < 0.05, ** *p* < 0.01 vs. Sham rats.

**Table 2 ijms-23-07666-t002:** Physiological and biochemical parameters in control and hyperbilirubinemia (HB) rats.

Parameters	Control Rats	HB Rats
Body weight (BW) (g)	241.2 ± 8.5	237.3 ± 10.7
Liver weight (%BW)	2.70 ± 0.48	3.31 ± 0.43 *
Spleen weight (%BW)	0.23 ± 0.05	0.39 ± 0.06 **
ALT (IU/L)	8.70 ± 4.70	10.08 ± 4.58
AST (IU/L)	16.69 ± 4.42	17.69 ± 3.31
ALP (IU/L)	116.93 ± 24.89	111.47 ± 44.67
Total bilirubin (µmol/L)	ND ^a^	12.32 ± 1.23 **
Total bile acids (µmol/L)	43.63 ± 10.82	37.29 ± 8.04

^a^, ND: <3.4 µmol/L. Data are expressed as mean ± S.D. (*n* = 6). * *p* < 0.05, ** *p* < 0.01 vs. control rats.

**Table 3 ijms-23-07666-t003:** Primer sequences for qRT-PCR for indicated genes in rat (r) and human (h).

Genes	Forward Primer (5′-3′)	Reverse Primer (5′-3′)
*rAbcb1a*	CAACCAGCATTCTCCATAATA	CCCAAGGATCAGGAACAATA
*rAbcb1b*	CTCGCTGCTATCATCCACGGAAC	CGCTGACGGTCTGTGTACTGTTG
*rAbcg2*	GTACTTTGCATCAGCAGGTTACCACT	ATTACAGCCGAAGAATCTCCGTTG
*rAbcc1*	GGCCTACTGAAGAGCAAGAC	GGATGATGATGACAGCTCC
*rAbcc2*	GTGTTTCCACAGAGCGGCTAG	GCTAGGCTGATATCAAGGAG
*rAbcc3*	ACACCGAGCCAGCCATATAC	ACATTGGCTCCGATAGCAAC
*rAbcc4*	TCTGGGTGGAAATCGGAATC	GCAGAATAACCAGAATGGCCA
*rAbcc5*	TGGAGAACGGGGACAACTT	AAAGGCGAGGTTTCAGCAG
*rAbcc6*	GCCACTACACCTGCCTTGACTCACCA	TCATGGGTGCTATTAGGGCGGATCAA
*rActb*	GCTATGTTGCCCTAGACTTCG	GCCACAGGATTCCATACCCAG
*hABCB1*	ACAGAGGGGATGGTCAGTGT	TCACGGCCATAGCGAATGTT
*hABCG2*	AACCTGGTCTCAACGCCATC	GTCGCGGTGCTCCATTTATC
*hABCC1*	CTACCTCCTGTGGCTGAATCTG	CATCAGCTTGATCCGATTGTCT
*hABCC2*	ATTCAGACGACCATCCAAAACGAGTT	GCCATAAAGTAAAAGGGTCCAGGGAT
*hABCC3*	CGCACACCGGCTTAACACTATCATGG	AAACCAGGAAAGGCCAGGAGGAAATC
*hABCC4*	GGATCCAAGAACTGATGAGTTAAT	TCACAGTGCTGTCTCGAAAATAG
*hABCC5*	ACCGCAGTCGTCGCACAGTCTCTCTC	GCGGGAACACACCAAACCACACAGCA
*hABCC6*	GCCTGCAGTGCTGGAGATGGAAGTGA	GTCCTTCCGGCTCTGATGCTCTGTGA
*hACTB*	AAGAGCTACGAGCTGCCTGAC	TCCTGCTTGCTGATCCACAT

## Data Availability

The data presented in this study are all contained within the main body of this article.

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
