# Peer review of "Bile Duct Ligation Impairs Function and Expression of Mrp1 at Rat Blood–Retinal Barrier via Bilirubin-Induced P38 MAPK Pathway Activations"

_ijms, 2022, doi:10.3390/ijms23147666_

Round 1
Reviewer 1 Report
Dr. Li et al have investigated the effect of the bile duct ligation (BDL) on the Mrp1 expression in the blood-retinal barrier and have attempted for elucidating the regulation pathway in rats. The research suggested that the activation of the P38 MAPK pathway induced by elevated bilirubin levels may contribute to Mrp1 impairment by the BDL, and the study contains interesting results that helps deepen the physiological function of Blood-retinal barrier. The reviewer have the following concerns and would appreciate it if authors could address them.
Figure 1: Although fluorescein and DNP-SG are Mrp1 substrates, the relation of these compounds between the concentrations in the retina and that in the plasma was different. In the retina, the fluorescein concentrations were higher in BDL rats, while the DNP-SG concentrations were similar in sham and BDL rats (in Figure 1a vs. 1d). In contrast, for the plasma, the fluorescein concentrations were similar, while the DNP-SG concentrations were lower in BDL rats (in Figure 1b vs. 1e). Why do the same Mrp1 substrates have different relation in the tissue concentrations? The reviewer couldn't understand. Please explain the reasons and add any evidence that the experimental data or the references. Why do the same Mrp1 substrates have different relation in the tissue concentrations? The reviewer couldn't understand. Please explain the reasons and add any evidence that the experimental data or the references.
Figure 1: Although authors have used UCB for an endogenous substrate, it is possible that the intra-retinal concentration also increased sharply just to maintain equilibrium due to the rapid increase in plasma concentration about 4 times. Is it possible to say that the rapid increase in UCB caused by the BDL treatment maintains the complete barrier function against membrane for UCB from plasma to the retina by MRP1? If there is a driving force with a concentration gradient that exceeds the efflux ability of MRP1, this investigation cannot draw any conclusions about the alteration of MRP1 function by UCB.
Figure 3: In the study using ARPE-19 cell monolayer, the concentration of substrates (UCB, NH4Cl, ADMA, and BAs) were different. What was the evidence for determining the concentration? Is it based on the concentrations obtained from the rat treated by BDL in the present study or any clinical evidence?
Author Response
Dear reviewer:
We have uploaded the revised manuscript entitled “Bile Duct Ligation Impairs Function and Expression of Mrp1 at Rat Blood-Retinal Barrier via Bilirubin-induced P38 MAPK Pathway Activations” (Manuscript ID: ijms-1739272). Thank you for your kind suggestions. We have done the revision according to your suggestions and highlighted the changes in our manuscript using red text for the revision. Below we have given an itemized list of responses to all the comments and questions you have raised.
We will be grateful to you if the manuscript can be reviewed for publication in “International Journal of Molecular Sciences (IJMS)”.
Thank you for your consideration.
Yours Sincerely,
Li Liu

Reviewer 2 Report
This is a very interesting manuscript about the role of Mrp1 at Rat Blood-Retinal Barrier via Bilirubin-induced P38 MAPK Pathway Activations. Results are clearly presented and supported by several in vitro - ex vivo and in vivo experiments. However, i have a concern about the significance of this well-explained adaptation mechanism in the onset of these pathologies. Does this mechanism has the role to retain bilirubin inside cells in order to improve antioxidant defenses?
1) authors can measure ROS in control and treated cells to verify if oxidative effects has a role in MRP1 reduction.
2) Once verified the above point, authors can furnish a rationale in the induced reduction of MRP1, aimed to keep bilirubin in cells to help its antioxidant function. In merit, authors can discuss and cite the following publications:
- In Antioxidants 2022, 11(6), 1072; https://doi.org/10.3390/antiox11061072 authors discuss on the pathogenesis of diabetic retinopathy, which often arises from functional alterations of the blood-retinal barrier (BRB) due to damaging oxidative stress reactions in lipids, proteins, and DNA. Authors showed that the treatment with tha antioxidant DHA of ARPE cells under high-glucose conditions activated erythroid 2-related factor Nrf2, which orchestrates the activation of cellular antioxidant pathways including HO1, which can have as an effect.
-Bilirubin concentration increases as higher HO-1 expression increases (Diabetes care 31, 1615–1620 (2008)).
-In Sci Rep. 2017; 7: 41681 authors found that a negative association between bilirubin concentration and the risk of diabetic retinopathy and diabetic neuropathy, indicating that bilirubin may play a protective role in the occurrence of diabetic complications.
Author Response
Dear reviewer:
We have uploaded the revised manuscript entitled “Bile Duct Ligation Impairs Function and Expression of Mrp1 at Rat Blood-Retinal Barrier via Bilirubin-induced P38 MAPK Pathway Activations” (Manuscript ID: ijms-1739272). Thank you for your kind suggestions. We have done the revision according to your suggestions and highlighted the changes in our manuscript using red text for the revision. Below we have given an itemized list of responses for all the comments and questions you have raised.
We will be grateful to you if the manuscript can be reviewed for publication in “International Journal of Molecular Sciences (IJMS)”.
Thank you for your consideration.
Yours Sincerely,
Li Liu

Round 2
Reviewer 1 Report
Since authors have addressed reviewer’s concerns one by one, the interpretation of the results presented this study became clarify.
Reviewer 2 Report
The authors adressed the comments.